# Gga-miR-30c-5p Enhances Apoptosis in Fowl Adenovirus Serotype 4-Infected Leghorn Male Hepatocellular Cells and Facilitates Viral Replication through Myeloid Cell Leukemia-1

**DOI:** 10.3390/v14050990

**Published:** 2022-05-07

**Authors:** Areayi Haiyilati, Linyi Zhou, Jiaxin Li, Wei Li, Li Gao, Hong Cao, Yongqiang Wang, Xiaoqi Li, Shijun J. Zheng

**Affiliations:** 1Key Laboratory of Animal Epidemiology of the Ministry of Agriculture, China Agricultural University, Beijing 100193, China; aray713@cau.edu.cn (A.H.); zlyi123321@126.com (L.Z.); lijiaxinswq@163.com (J.L.); liwei19940506@163.com (W.L.); gaoli194@cau.edu.cn (L.G.); caohong@edu.cau.cn (H.C.); vetwyq@cau.edu.cn (Y.W.); 2College of Veterinary Medicine, China Agricultural University, Beijing 100193, China

**Keywords:** FAdV-4, chicken microRNA, gga-miR-30c-5p, apoptosis, Mcl-1

## Abstract

Fowl adenovirus serotype 4 (FAdV-4) is the primary causative agent responsible for the hepatitis-hydropericardium syndrome (HHS) in chickens, leading to considerable economic losses to stakeholders. Although the pathogenesis of FAdV-4 infection has gained attention, the underlying molecular mechanism is still unknown. Here, we showed that the ectopic expression of gga-miR-30c-5p in leghorn male hepatocellular (LMH) cells enhanced apoptosis in FAdV-4-infected LMH cells by directly targeting the myeloid cell leukemia-1 (Mcl-1), facilitating viral replication. On the contrary, the inhibition of endogenous gga-miR-30c-5p markedly suppressed apoptosis and viral replication in LMH cells. Importantly, the overexpression of Mcl-1 inhibited gga-miR-30c-5p or FAdV-4-induced apoptosis in LMH cells, reducing FAdV-4 replication, while the knockdown of Mcl-1 by RNAi enhanced apoptosis in LMH cells. Furthermore, transfection of LMH cells with gga-miR-30c-5p mimics enhanced FAdV-4-induced apoptosis associated with increased cytochrome *c* release and caspase-3 activation. Thus, gga-miR-30c-5p enhances FAdV-4-induced apoptosis by directly targeting Mcl-1, a cellular anti-apoptotic protein, facilitating FAdV-4 replication in host cells. These findings could help to unravel the mechanism of how a host responds against FAdV-4 infection at an RNA level.

## 1. Introduction

Fowl adenoviruses (FAdVs), belonging to the genus *Aviadenovirus*, the family of *Adenoviridae*, are generally divided into five species (*FAdV-A* to *FAdV-E*) based on their restriction enzyme digestion patterns [1], including 12 serotypes (FAdV1-7, FAdV8a/b, and FAdV9-11) as determined by a serum cross-neutralization test [2]. Although FAdVs could be isolated from healthy chickens, some serotypes of FAdVs were associated with several notable diseases, such as inclusion body hepatitis (IBH) [3], hepatitis-hydropericardium syndrome (HHS) [4], and adenoviral gizzard erosion (AGE) [5] in chickens and other birds. FAdV-4 is the primary causative agent responsible for HHS at mortality of 30–80% [6,7,8]. The predominant gross lesions of HHS in chicken are hydropericardium, accumulation of clear or yellowish jelly-like fluid in the pericardial sac, as well as a yellow brown-colored and swollen, enlarged, congested, and friable liver with foci of hemorrhages and necrosis, and in a few cases, hemorrhages in spleens, lungs, and kidneys [9,10], and enlarged bursa of Fabricius could be observed [11]. In the past few years, the occurrence of HHS in Asia, particularly in China, has gained attention, leading to considerable economic losses to stakeholders [12]. FAdVs, highly resistant to inactivation, persistently exist in the environment for a long period of time and can be transmitted both horizontally and vertically [13]. Once flocks were infected with FAdVs, the virus would circulate in flocks for a long time. Thus, FAdVs pose severe threats to the poultry industry around the world. 

FAdV-4, a non-enveloped and double-stranded DNA (dsDNA) virus, potentially contains 46 ORFs encoding 11 structural proteins and approximately 32 non-structural proteins [14]. Among these viral proteins, Hexon, a structural protein with a mass of 107 kD, is the most abundant capsid protein, containing group, type, and subtype-specific antigenic determinants [15,16] and determining the FAdV-4 pathogenicity [17]. It was found that the FAdV-4 Hexon interacted with chaperonin containing TCP-1 subunit eta (CCT7), enhancing viral replication [18]. PX, another structural protein of FAdV, is involved in attaching the linear double-stranded DNA genome to the capsid during replication [19] and induces apoptosis in FAdV-4-infected LMH cells, serving as a virulence factor for FAdV-4 infection [20]. In addition, other viral proteins were also involved in the pathogenesis of FAdV-4 infection [21].

MicroRNAs (miRNAs), small non-coding RNAs, are abundant classes of post-transcriptional regulators with the length of 20–24 nucleotides that are involved in the control of a broad range of cellular activities, such as cell proliferation [22], differentiation [23], apoptosis [24], and metabolism [25], and play a role in the replication and propagation of viruses [26,27]. The nucleotides 2–7 of the miRNA 5′ end, which is called the seed-sequence, usually target the 3′ untranslated region (3′-UTR), 5′ untranslated region (5′-UTR), or the coding sequence (CDS) of mRNAs either fully or partially to degrade mRNA or inhibit translation [28,29,30].

However, the role of miRNAs in the host response to FAdV-4 infection remains elusive. In the present study, we found that the infection of LMH cells with FAdV-4 altered the expressions of miRNAs in host cells, among which gga-miR-30c-5p expression was downregulated and attracted our attention. We found that the transfection of host cells with gga-miR-30c-5p enhanced apoptosis in FAdV-4-infected LMH cells by directly targeting the myeloid cell leukemia-1 (Mcl-1), a cellular anti-apoptotic protein, favoring FAdV-4 growth in LMH cells, while the inhibition of gga-miR-30c-5p in FAdV-4-infected cells slows down viral replication. These data indicate that gga-miR-30c-5p serves as a pro-apoptotic factor in the host response to FAdV-4 infection, uncovering a novel mechanism of host response against FAdV-4 infection at an RNA level. 

## 2. Materials and Methods

### 2.1. Cells and Virus

LMH cells, an immortalized chicken liver cell line, were kindly provided by Dr. Jinhua Liu (CAU, Beijing, China). The LMH cells were cultured in Waymouth MB 752/1 Medium (MACGENE Technology, Beijing, China) supplemented with 1× Penicillin-Streptomycin (MACGENE Technology) and 10% fetal bovine serum (Gibco, Grand Island, NE, USA) in a 5% CO_2_ incubator at 37 °C. The cell culture plates were coated with 0.1% gelatin solution (Cat. ES-006-B, Millipore, Billerica, MA, USA) and incubated at 4 °C for 10 min before splitting cells. The FAdV4 HuBWH strain was originally isolated from the liver of a diseased chicken [18] and was stocked at −80 °C till use.

### 2.2. Reagents, Chemicals, and Antibodies

OPTI-MEM™ I and Lipofectamine 3000 transfection reagents were purchased from Invitrogen (Carlsbad, CA, USA). The jetPRIMETM transfection reagent was obtained from Polyplus-transfection Biotechnology Company (Strasbourg, France). Mouse anti-GAPDH antibodies (60004-1-Ig) were purchased from Proteintech (Wuhan, China), anti-Tubulin antibodies from Abcam (Cambridge, UK), anti-Hexon monoclonal antibodies and anti-PX monoclonal antibodies from CAEU Biological Company (Beijing, China), anti-Mcl-1 monoclonal antibodies (ab32087) and anti-cytochrome *c* antibodies (ab133504) from Abcam (Cambridge, UK), and horseradish peroxidase (HRP)-conjugated goat anti-mouse/rabbit IgG antibodies from DingGuoShengWu (Beijing, China). An annexin V-PE/7-AAD apoptosis detection kit was purchased from BD Pharmingen (Franklin Lakes, NJ, USA). A dual-specific luciferase assay kit was purchased from 122 Promega (Madison, WI, USA). Caspase-3 colorimetric assay kits were obtained from BioVision (San Francisco, CA, USA). The pRK5-Flag plasmid was obtained from Clontech (Mountain View, CA, USA). The CCK-8 solution was purchased from Beyotime Biotechnology (Beijing, China).

### 2.3. Plasmid Construction

Gallus gallus Mcl-1 (GenBank accession No. NM_001257283.4) was cloned from LMH cells with the primer pairs shown in Table 1. The primers were synthesized by Sangon Company (Shanghai, China). pRK5-Flag-Mcl-1 expression plasmids were constructed by standard molecular biology techniques.

### 2.4. Sequences of miRNA Mimics or Inhibitors

Mimics/inhibitors for miRNA were synthesized by Genepharma Company (Shanghai, China). The sense sequences for miRNAs are shown in Table 2.

### 2.5. miRNA Target Prediction 

The miRNA targets in host cells were predicted by RNA22.v2 (https://cm.jefferson.edu/rna22, accessed on 15 January 2022), miRanda (http://www.microrna.org/microrna/home.do, accessed on 15 January 2022), and Targetscan (http://www.targetscan.org/vert_70/, accessed on 15 January 2022).

### 2.6. RNA Isolation and Quantitative Real-Time PCR (qRT-PCR) Analysis

Total RNA and miRNA were prepared from LMH cells using the EASYspin Plus kit and RNA misi microRNA kit (Aidlab, Beijing, China), respectively, as per the manufacturer’s instructions. Quantitative reverse transcription-PCR (qRT-PCR) was performed using a PrimeScript RT reagent kit (TaKaRa) on a Light Cycler 480 II (Roche, Basel, Switzerland). The primers used for qRT-PCR are shown in Table 3. All primers were designed and synthesized by Sangon Company. The thermal cycling parameters were as follows: 94 °C for 2 min; 40 cycles of 94 °C for 20 s, 56 °C for 20 s, and 72 °C for 20 s; and 1 cycle of 95 °C for 30 s, 60 °C for 30 s, and 95 °C for 30 s. qRT-PCR analysis of gga-miR-30c-5p was performed with an RT-PCR Quantitation Kit (GenePharma, Suzhou, China). The thermal cycling parameters for miRNA were as follows: 95 °C for 3 min; 40 cycles of 95 °C for 12 s, 62 °C for 40 s; and 1 cycle of 95 °C for 30 s, 60 °C for 30 s, and 95 °C for 30 s. The final step was to obtain a melt curve for the PCR products to determine the specificity of the amplification, and the U6 snRNA was utilized as the reference gene. The expression levels of gga-miR-30c-5p were calculated relative to that of U6 snRNA and presented as fold increases or decreases relative to the control samples. All samples were carried out in triplicate on the same plate.

### 2.7. Apoptosis Assay

LMH cells were seeded on 12-well plates and cultured for 24 h, followed by transfection with miRNA controls, gga-miR-30c-5p mimics, or gga-miR-30c-5p inhibitors. At different time points (24 and 48 h) post-transfection, cells were harvested, double stained with 7-AAD and Annexin V-PE using an apoptosis detection kit (BD Pharmingen™), and examined by flow cytometry. To determine the effect of gga-miR-30c-5p on FAdV-4-induced apoptosis, LMH cells were cultured and transfected with miRNA controls, gga-miR-30c-5p mimics, or gga-miR-30c-5p inhibitors. Twenty-four hours after transfection, the cells were infected with FAdV-4 at an MOI of 1 and harvested at different time points (24 and 48 h) post-infection for further analysis. To confirm the effect of Mcl-1 on cell apoptosis, LMH cells were transfected with pRK5-Flag-Mcl-1 or empty vectors. Twenty-four hours after transfection, the cells were harvested. Similarly, LMH cells were seeded onto 12-well plates and cultured for 24 h before transfection with siRNA against Mcl-1 or RNAi controls using Lipofectamine 3000. Double transfections were performed at 24 h intervals. Twenty-four hours after the second transfection, cells were harvested for further analysis. The samples were subjected to flow cytometry analysis as described above. The data were then analyzed with CellQuest Pro software (version 5.1, BD) and presented as means ± standard deviations (SD) of three independent experiments.

### 2.8. Measurement of FAdV-4 Growth in LMH Cells

Normal cells or cells receiving gga-miR-30c-5p mimics, miRNA controls (80 nM per well), gga-miR-30c-5p inhibitors, or miRNA inhibitor controls (200 nM per well) were infected with FAdV-4 at an MOI of 1. Cell culture samples were harvested at different time points (12, 24, 48, and 72 h) post-FAdV-4 infection. To confirm the effect of Mcl-1 on FAdV-4 replication, LMH cells were transfected with pRK5-Flag-Mcl-1 or an empty vector. Twenty-four hours after transfection, the cells were infected with FAdV-4 at an MOI of 1 and harvested after 24 h. The samples were subjected to three rounds of freeze-thawed treatment. The viral contents in the total cell lysates were titrated using 50% tissue culture infective doses (TCID_50_) in LMH cells. Briefly, the viral solution was diluted 10-fold in Waymouth MB 752/1 Medium. A 100 μL aliquot of each diluted sample was added to the wells of 96-well plates, followed by the addition of 100 μL of LMH cells at a density of 3 × 10^5^ cells/mL. The cells were cultured at 37 °C in 5% CO_2_ for 5 days. Tissue culture wells with cytopathic effect (CPE) were determined as positive. The titer was calculated based on a previously described method [31].

### 2.9. Western Blot Analysis

All cell lysates were prepared using a lysis buffer (50 mM Tris-HCl, pH 8.0, 150 mM NaCl, 5 mM EDTA, 1% NP-40, 10% glycerol, 1× complete cocktail protease inhibitor), boiled with SDS loading buffer for 10 min, and fractionated by electrophoresis on 10% SDS-PAGE gels. The resolved proteins were then transferred onto polyvinylidene difluoride (PVDF) membranes. After being blocked with 5% skimmed milk, the membranes were incubated with anti-Hexon, anti-PX, anti-Mcl-1, anti-cytochrome *c*, anti-GAPDH, or anti-Tubulin antibodies, followed by incubation with HRP-conjugated secondary antibodies. Western Blots were developed using an enhanced chemiluminescence (ECL) kit (Kangwei Biological Company, Suzhou, China) per the manufacturer’s instructions.

### 2.10. Luciferase Reporter Gene Assays

LMH cells were seeded on 24-well plates and cultured overnight, followed by transfection with luciferase reporter gene plasmids (pGL3-target-Mcl-1-wt or pGL3-target-Mcl-1-mutant) and miRNA mimics, miRNA controls, miRNA inhibitors, or miRNA inhibitor controls. To normalize for transfection efficiency, another plasmid pRL-TK expressing Renilla luciferase reporter gene was added to each transfection as a control. Forty-eight hours post-transfection, luciferase reporter gene assays were performed with a dual-luciferase reporter gene assay system. Firefly luciferase activities were normalized on the basis of Renilla luciferase activities.

### 2.11. Knockdown of Mcl-1 by RNAi

The small interfering RNAs (siRNAs) were designed and synthesized by Genepharma Company (Suzhou, China) and used to knock down the expression of Mcl-1 in LMH cells. The sequences of siRNA against Mcl-1 in LMH cells included RNAi#1 and RNAi#2 and are shown in Table 4. LMH cells were seeded onto 12-well plates and cultured for 24 h before transfection with siRNA or controls using Lipofectamine 3000. Double transfections were performed at 24 h intervals. Twenty-four hours after the second transfection, cells were harvested for further analysis.

### 2.12. Measurement of Cytochrome C Release

The isolation of mitochondria and cytosol was performed using a cell mitochondrion isolation kit (Beyotime Biotechnology, Beijing, China). Briefly, LMH cells were transfected with miRNAs for 24 h, followed by mock infection or infection with FAdV-4 at an MOI of 1 for 24 and 48 h. Untreated cells or cells receiving miRNAs were incubated in 100 μL of ice-cold mitochondrion lysis buffer on ice for 15 min, and cell suspensions were homogenized on ice with a Dounce grinder. The homogenates were centrifuged at 600× *g* for 10 min at 4 °C, and the supernatant was obtained and then centrifuged again at 12,000× *g* for 20 min at 4 °C. The supernatant was examined for cytochrome *c* release by Western Blot using anti-cytochrome *c* antibodies.

### 2.13. Caspase-3 Activity Assays

LMH cells were seeded on 6-well plates before being transfected with gga-miR-30c-5p mimics, miRNA controls, gga-miR-30c-5p inhibitors, or miRNA inhibitor controls. Twenty-four hours after transfection, cells were mock-infected or infected with FAdV-4 at an MOI of 1. Twenty-four or forty-eight hours after infection, cell lysates were prepared and examined for caspase-3 activities using caspase-3 activity assay kits per the manufacturer’s instructions. The samples were measured at 405 nm with a microplate reader (Tecan: Sunrise) using the fluorescent substrate DEVD-pNA (synthetic caspase-3 substrate). 

### 2.14. Cell Viability Assay

The LMH cells were seeded on 96-well culture plates and transfected with Mcl-1-RNAi#1 or control-RNAi. At different time points (24, 48, and 72 h) post-transfection, 10 μL of CCK-8 solution (Beyotime Biotechnology, China) was added to each well, followed by incubation at 37 °C for 1 h, and the absorbance of the solution was finally determined at 450 nm using a microplate spectrophotometer. The results are representative of three independent experiments.

### 2.15. Statistical Analysis

The statistical analysis was performed using GraphPad Prism version 9.0. The significance of the differences between FAdV-4 infection and mock controls and between gga-miR-30c-5p mimics/inhibitors and miRNA controls in gene expression, apoptosis, cytochrome *c* release, caspase activity, cell viability and viral growth was determined by a Mann–Whitney test or analysis of variance (ANOVA) accordingly.

## 3. Results

### 3.1. Infection of LMH Cells with FAdV-4 Reduced gga-miR-30c-5p Expression

To explore the roles of miRNA in the host response against FAdV-4 infection, we performed deep sequencing to analyze the miRNA expressions in LMH cells infected with the FAdV-4 HuBWH strain at an MOI of 1 for 24 h. As shown in Figure 1A, 67 miRNAs were encoded by the FAdV-4 DNA genome, 552 miRNAs were found changed in LMH cells after FAdV-4 infection, while 132 were unchanged in LMH cells. To explore the role of miRNAs in the host response to FAdV-4 infection, we selected 12 miRNAs with altered expression levels and examined their effects on host cell apoptosis and virus replication. Among them, gga-miR-30c-5p was apparently decreased after FAdV-4 infection (Figure 1B), attracting our attention. It was reported that gga-miR-30c-5p inhibited glioma proliferation and invasion via targeting Bcl-2 [32] and regulated cisplatin-induced apoptosis of renal tubular epithelial cells via targeting Bnip3L and Hspa5 [33]. Thus, we set out to examine the role of gga-miR-30c-5p in the host response to FAdV-4 infection. We infected LMH cells with different doses of FAdV-4 and examined the expression of gga-miR-30c-5p at 24 h post-infection. As shown in Figure 1C, the expression of gga-miR-30c-5p markedly decreased in a dose-dependent manner 24 h post-FAdV-4 infection, indicating that gga-miR-30c-5p might play a role in host cell response to FAdV-4 infection.

### 3.2. Gga-miR-30c-5p Enhances Apoptosis in LMH Cells with or without FAdV-4 Infection

As we previously found that FAdV-4-induced apoptosis in LMH cells [20], it was intriguing to examine the effect of gga-miR-30c-5p on FAdV-4-induced apoptosis. We transfected LMH cells with gga-miR-30c-5p mimics or inhibitors and examined apoptosis in cells 24 and 48 h post-transfection using flow cytometry. As shown in Figure 2A,B, the cells receiving gga-miR-30c-5p displayed marked apoptosis compared to that of miRNA controls (*p* < 0.01), while the inhibition of endogenous gga-miR-30c-5p with its inhibitors significantly inhibited apoptosis (*p* < 0.05), indicating that gga-miR-30c-5p is involved in spontaneous cell death in LMH cells. To explore the role of gga-miR-30c-5p in FAdV-4-induced apoptosis, we transfected LMH cells with gga-miR-30c-5p mimics, gga-miR-30c-5p inhibitors, or miRNA controls, and examined apoptosis 24 and 48 h post-FAdV-4 infection. As shown in Figure 2C,D, the overexpression of gga-miR-30c-5p markedly enhanced apoptosis in FAdV-4-infected LMH cells (*p* < 0.01). On the contrary, the inhibition of endogenous gga-miR-30c-5p significantly reduced apoptosis in FAdV-4-infected cells (*p* < 0.05). These data clearly indicate that gga-miR-30c-5p serves as a pro-apoptotic factor, enhancing apoptosis in host cells with or without FAdV-4 infection.

### 3.3. Gga-miR-30c-5p Promotes FAdV-4 Replication in LMH Cells

Since gga-miR-30c-5p promoted FAdV-4-induced apoptosis in host cells, we attempted to examine the effect of gga-miR-30c-5p on FAdV-4 replication. We transfected LMH cells with different doses of gga-miR-30c-5p and examined the expression of Hexon and PX, structural proteins of FAdV-4, in gga-miR-30c-5p-transfected cells post FAdV-4 infection by Western Blot using anti-Hexon and anti-PX monoclonal antibodies. As shown in Figure 3A–C, both Hexon and PX expressions increased in FAdV-4-infected cells with gga-miR-30c-5p transfection in a dose-dependent manner (*p* < 0.05), suggesting that gga-miR-30c-5p promotes FAdV-4 replication. Furthermore, we examined the viral titers in gga-miR-30c-5p-transfected LMH cells at different time points (12, 24, 48, and 72 h) post-FAdV-4 infection using TCID_50_ assay. As a result, the transfection of LMH cells with gga-miR-30c-5p markedly promoted FAdV-4 replication compared to that of control cells (*p* < 0.05) (Figure 3D). These results indicate that gga-miR-30c-5p promoted FAdV-4 replication in host cells. To consolidate the above results, we inhibited the expression of endogenous gga-miR-30c-5p in LMH cells by specific miRNA inhibitors and examined the viral titers in these cells. As shown in Figure 4A, the inhibition of endogenous gga-miR-30c-5p markedly reduced viral titers in FAdV-4-infected cells (*p* < 0.05). Consistently, the expressions of both Hexon and PX proteins significantly decreased in gga-miR-30c-5p inhibitor-treated cells post FAdV-4 infection compared with that of controls as examined by Western Blot assay (*p* < 0.05) (Figure 4B–D). These data clearly show that gga-miR-30c-5p promotes FAdV-4 replication in host cells.

### 3.4. The Mcl-1 Gene Is a Target of gga-miR-30c-5p in LMH Cells

Since gga-miR-30c-5p is involved in apoptosis, it is intriguing to explore the underlying molecular mechanism. Using the RNA22 (version 2) databases, we identified Mcl-1 as a target of gga-miR-30c-5p in host cells. The region of the Mcl-1 CDS at bp 874 to 895 contains the target site for gga-miR-30c-5p (Figure 5A). Mcl-1, an anti-apoptotic oncoprotein, is a member of the Bcl-2 family [34]. It was reported that human miR-30c-5p caused apoptosis via targeting apoptosis-related genes, such as Bcl-2 [32], and that some miRNAs induced apoptosis via targeting Mcl-1 [35,36]. Therefore, we hypothesized that gga-miR-30c-5p might regulate apoptosis by targeting Mcl-1 in FAdV-4-infected cells. To test this hypothesis, we constructed a firefly luciferase reporter gene plasmid (pGL3-Mcl-1-WT, where WT is wild type) containing the predicted target site in Mcl-1 and another construct (pGL3-Mcl-1-Mut) with mutations in the targeted regions as controls, transfected LMH cells with these reporter gene plasmids and miRNAs and performed luciferase reporter gene assays. As shown in Figure 5B, the transfection of cells with gga-miR-30c-5p together with pGL3-Mcl-1-WT significantly reduced luciferase activity of the reporter gene compared with that of controls (*p* < 0.01), but this reduction could be completely abolished by transfection with pGL3-Mcl-1-Mut, indicating that gga-miR-30c-5p inhibited Mcl-1 expression by targeting its specific sequence in the Mcl-1 gene CDS. On the contrary, the inhibition of endogenous gga-miR-30c-5p by miRNA inhibitors promoted luciferase activity of the reporter gene (*p* < 0.05), suggesting that Mcl-1 expression is enhanced in cells with a reduced level of gga-miR-30c-5p. Furthermore, we examined the effect of gga-miR-30c-5p on Mcl-1 mRNA expression by qRT-PCR. Consistently, our results show that the transfection of LMH cells with gga-miR-30c-5p decreased Mcl-1 mRNA expression compared with that of miRNA controls (*p* < 0.01), while the knockdown of endogenous gga-miR-30c-5p increased the mRNA expression of Mcl-1 (*p* < 0.001) (Figure 5C). These data indicate that gga-miR-30c-5p reduced Mcl-1 expression at an mRNA level. To consolidate these findings, we next examined the effect of gga-miR-30c-5p on Mcl-1 protein expression by Western Blot assay. As a result, the overexpression of gga-miR-30c-5p in LMH cells markedly suppressed Mcl-1 expression (*p* < 0.001) (Figure 5D,E), while the knockdown of endogenous gga-miR-30c-5p by its inhibitors enhanced Mcl-1 expression compared with that of miRNA inhibitor controls (*p* < 0.01) (Figure 5F,G). These data clearly show that gga-miR-30c-5p inhibits Mcl-1 expression by directly targeting Mcl-1 gene expression in LMH cells.

### 3.5. Mcl-1 Suppressed gga-miR-30c-5p-Induced Apoptosis in LMH Cells

The facts that gga-miR-30c-5p promoted apoptosis in LMH cells and that gga-miR-30c-5p inhibited Mcl-1 expression suggest that gga-miR-30c-5p might promote apoptosis by targeting Mcl-1 and that the overexpression of Mcl-1 in host cells would, therefore, block gga-miR-30c-5p-induced apoptosis while the knockdown of Mcl-1 would promote apoptosis as gga-miR-30c-5p did. To test this hypothesis, we transfected LMH cells with gga-miR-30c-5p alone or together with pRK5-Flag-Mcl-1 and examined the effect of overexpressed Mcl-1 on gga-miR-30c-5p-induced apoptosis by flow cytometry. As shown in Figure 6A,B, the overexpression of Mcl-1 markedly blocked gga-miR-30c-5p-induced apoptosis in LMH (*p* < 0.01), suggesting that Mcl-1 is an anti-apoptotic molecule, suppressing miR-30c-5p-induced apoptosis. Furthermore, we made two Mcl-1 RNAi#1 and RNAi#2 constructs and found that both constructs could effectively lower the cellular level of Mcl-1 (Figure 6C). As shown in Figure 6D, transfection with Mcl-1 RNAi#1 in cells decreased the viability and proliferation of cells compared to that of control RNAi, indicating that the knockdown of endogenous Mcl-1 suppressed the viability and proliferation of LMH cells. We then examined apoptosis in LMH cells receiving the Mcl-1 RNAi#1 or control RNAi. As a result, the knockdown of Mcl-1 by RNAi markedly promoted apoptosis in LMH cells as gga-miR-30c-5p did (*p* < 0.001) (Figure 6E,F), indicating that Mcl-1 serves as an anti-apoptotic factor in host cells and that gga-miR-30c-5p induces apoptosis in cells by targeting Mcl-1.

### 3.6. Mcl-1 Is Involved in FAdV-4-Induced Apoptosis and Suppressed FAdV-4 Replication

Since gga-miR-30c-5p promoted FAdV-4-induced apoptosis and viral replication by targeting Mcl-1, it is likely that Mcl-1 is involved in FAdV-4-induced apoptosis and affects viral replication. Thus, we transfected the LMH cells with pRK5-Flag-Mcl-1 or pRK5-Flag as a control and examined apoptosis and viral replication in LMH cells post FAdV-4 infection. As a result, the overexpression of Mcl-1 in FAdV-4-infected cells markedly reduced FAdV-4-induced apoptosis in host cells (*p* < 0.01) (Figure 7A,B), accompanied by reduced expression of viral proteins Hexon and PX in FAdV-4-infected cells (*p* < 0.01) (Figure 7C–E), indicating that Mcl-1 suppressed FAdV-4-mediated apoptosis, suppressing viral infection. To confirm the results, we transfected LMH cells with pRK5-Flag-Mcl-1 or pRK5-Flag and examined the viral titers in these cell cultures 24-h-post FAdV-4 infection using TCID_50_ assay. Consistently, the transfection of LMH cells with pRK5-Flag-Mcl-1 significantly inhibited FAdV-4 replication compared to that of controls (*p* < 0.01) (Figure 7F). These data indicate that Mcl-1 acts as an anti-apoptotic factor in FAdV-4-infected cells and suppresses FAdV-4 replication.

### 3.7. Overexpression of Gga-miR-30c-5p Enhanced FAdV-4-Induced Cytochrome C Release and Activation of Caspase-3

As one of the vital anti-apoptotic molecules of the Bcl-2 family, Mcl-1 interferes in a cascade of events leading to the release of cytochrome *c* from mitochondria to promote cell survival [37]. Since gga-miR-30c-5p inhibits the expression of Mcl-1 and Mcl-1-mediated apoptosis is associated with cytochrome *c* release, we hypothesized that gga-miR-30c-5p enhanced apoptosis via facilitating cytochrome *c* release and the activation of caspase-3. To test this hypothesis, we transfected LMH cells with the miRNAs or miRNA controls and examined cytochrome *c* release and caspase-3 activities 24 and 48 h post-FAdV-4 infection. As shown in Figure 8A,B, the transfection of LMH cells with gga-miR-30c-5p significantly increased cytochrome *c* release compared to that of controls (*p* < 0.01). Similarly, the overexpression of gga-miR-30c-5p significantly increased caspase-3 activity (*p* < 0.01), while the inhibition of endogenous gga-miR-30c-5p by inhibitors suppressed the activity of caspase-3 (Figure 8C,D). These data demonstrate that gga-miR-30c-5p enhances apoptosis via facilitating cytochrome *c* release and activation of caspase-3.

Taken together, our data show that the infection of LMH cells with FAdV-4 decreased gga-miR-30c-5p expression. Importantly, gga-miR-30c-5p enhanced FAdV-4-induced apoptosis in LMH cells by directly targeting Mcl-1, facilitating viral replication in host cells. The overexpression of Mcl-1 inhibited gga-miR-30c-5p or FAdV-4-induced apoptosis in LMH cells, reducing FAdV-4 replication, while the knockdown of Mcl-1 by RNAi enhanced apoptosis in LMH cells. Furthermore, gga-miR-30c-5p enhanced FAdV-4-induced apoptosis associated with increased cytochrome *c* release and caspase-3 activation. Thus, gga-miR-30c-5p enhances apoptosis in FAdV-4-infected cells by directly targeting Mcl-1, a cellular anti-apoptotic protein, facilitating FAdV-4 replication in host cells.

## 4. Discussion

FAdV-4 is one of the most important pathogens for chickens and was first identified as a causative agent responsible for HHS in three to six-week-old broilers in Pakistan in 1987 [38], and subsequently, HHS occurred around the world [8,39,40,41]. In contrast to mild disease caused by other FAdV serotypes, the outbreaks of HHS in chickens infected by FAdV-4 caused tremendous economic losses to the poultry industry in China in 2015. Thus, effective control of HHS by vaccination is crucial to the income of stakeholders. A complete understanding of the mechanisms underlying FAdV-induced pathogenesis is required for the development of highly efficacious novel vaccines against HHS. However, currently, the pathogenesis of FAdV infection is still largely unknown. In the present study, first, our data show that the infection of LMH cells decreased gga-miR-30c-5p expression. Second, the overexpression of gga-miR-30c-5p significantly enhanced FAdV-induced apoptosis and facilitated viral replication in LMH cells, while the knockdown of gga-miR-30c-5p by specific inhibitors reduced apoptosis and suppressed viral replication in FAdV-infected cells. Thirdly, and importantly, the overexpression of gga-miR-30c-5p in LMH cells induced apoptosis by directly targeting cellular anti-apoptotic factor Mcl-1, while this induction could be blocked by overexpressed Mcl-1. Fourth, the knockdown of Mcl-1 by RNAi induced apoptosis just as the overexpression of gga-miR-30c-5p did. Finally, gga-miR-30c-5p-enhanced apoptosis in FAdV-4-infected cells was associated with increased cytochrome *c* release and caspase-3 activation. Thus, gga-miR-30c-5p enhances FAdV-4-induced apoptosis by directly targeting Mcl-1 and facilitates FAdV-4 replication in host cells, highlighting a critical role of miRNAs in host defense against FAdV-4 infection.

It is well known that miRNAs, small and non-coding RNA molecules, negatively regulate gene expression by binding to mRNAs and thereby promoting degradation of the target mRNA or blocking its translation into protein [42]. Studies about the downregulating expression of miRNAs post-viral infection are relatively rare. In our study, FAdV-4 infection with LMH cells reduced the expression of gga-miR-30c-5p, and the overexpression of gga-miR-30c-5p promoted the replication of FAdV-4. FAdV-4 infection triggered host response against viral infection by downregulating gga-miR-30c-5p expression. It has been reported that VSV decreased miR-27a in different immune cells and found that the IFN/JAK/STAT1 signal pathway induced downregulation of downstream miR-27a by RUNX1, which was inhibited by VSV [43]. Therefore, we will focus on the mechanism of regulating gga-miR-30c-5p by FAdV-4 in the future. miRNAs play important roles in host responses to pathogenic infection, while some viruses have evolved with the capacity of manipulating host miRNAs to evade host response to benefit viral replication [44,45]. For example, gga-miR-181a-5p facilitated FAdV-4 replication via targeting of STING [46], miR-30c promoted type 2 PRRSV infection by targeting the interferon–alpha/beta receptor beta chain [47], gga-miR-142-5p attenuated IRF7 signaling and promoted replication of IBDV by targeting the chMDA5′s 3′ untranslated region [48], and gga-miR-16c-5p induced apoptosis in IBDV-infected cells via targeting Bcl-2, facilitating viral replication [26]. In the present study, we found that gga-miR-30c-5p enhanced FAdV-4-induced apoptosis by directly targeting Mcl-1, promoting FAdV-4 replication in host cells. Thus, it seems that cellular miRNAs could not only serve as components of host response in maintaining homeostasis but also be taken advantage of by pathogens for their own survival. In humans, miR-30c-5p was involved in apoptosis, inflammation, autophagy, and other physiological processes by different targets, including ATG5 [49], eIF2α [50], SIRT1 [51], and FOXO3 [52]. As far as we know, this study is the first report to show that gga-miR-30c-5p is involved in FAdV-4-induced apoptosis in host cells by targeting Mcl-1, an anti-apoptotic member of the Bcl-2 family. 

Apoptosis is a strictly controlled physiological process that eliminates unwanted or potentially harmful cells. We previously reported that some miRNAs were involved in apoptosis in host cells, affecting viral replication, for instance, gga-miR-16c-5p enhanced IBDV-induced apoptosis and viral replication by targeting Bcl-2 [26], and gga-miR-29a-3p suppressed avian reovirus-induced apoptosis and viral replication via targeting caspase-3 [53]. In the present study, our results show that gga-miR-30c-5p enhanced apoptosis in FAdV-4-infected cells by targeting Mcl-1, facilitating viral replication. Furthermore, our previous study demonstrated that PX of FAdV-4 acts as a major viral factor inducing apoptosis in FAdV-4-infected LMH cells [20]. It is highly possible that FAdV-4-induced apoptosis is attributable to multiple factors at both protein and RNA levels. Interestingly, we noted that all these viruses (IBDV, Reovirus, and FAdV) are non-enveloped viruses; apoptosis may be one of the important means required for their release and spread. Thus, from an evolutionary point of view, it is very important for FAdV-4 to have the capability of inducing apoptosis in host cells at both the protein and RNA levels.

Mcl-1, a member of the Bcl-2 anti-apoptotic protein subfamily, was first identified in the ML-1 human myeloid leukemia cell line in 1993 [54]. Mcl-1 was expressed differentially in all cells, especially in cancer cells, and its expression was downregulated during apoptosis in many cell types in contrast to the anti-apoptotic Bcl-2 and Bcl-X_L_ proteins [55,56]. It has been reported that several miRNAs, such as miR-101 [57], miR-519d [58], miR-1469 [35], and miR-26b [59], play a role in apoptosis by targeting Mcl-1. In this study, we found that gga-miR-30c-5p enhanced apoptosis in FAdV-4-infected cells by targeting Mcl-1, facilitating viral replication. Of note, there are several questions to be addressed: for example, (1) what is the mechanism underlying FAdV4-induced gga-miR-30c-5p expression? (2) Is the cellular sensor, such as c-GAS or ALRs (AIM-2-like receptors), involved in the recognition of genomic DNA of FAdV-4 to initiate host response? (3) If so, which transcription factors are involved in these biological processes? More efforts will be required to elucidate the molecular mechanisms of FAdV-4-induced pathogenesis.

In conclusion, our data show that the infection of LMH cells with FAdV-4 altered gga-miR-30c-5p expression. Importantly, gga-miR-30c-5p enhanced apoptosis in FAdV-4-infected cells by directly targeting Mcl-1, facilitating viral replication. Furthermore, gga-miR-30c-5p enhanced FAdV-4-induced apoptosis associated with increased cytochrome *c* release and caspase-3 activation. Thus, gga-miR-30c-5p plays an important role in FAdV-4-induced apoptosis, favoring FAdV-4 survival in host cells, which might be used as a potential target for intervening FAdV-4 infection. These findings contribute to the understanding of the mechanisms underlying FAdV-4-induced apoptosis at an RNA level.

## Figures and Tables

**Figure 1 viruses-14-00990-f001:**
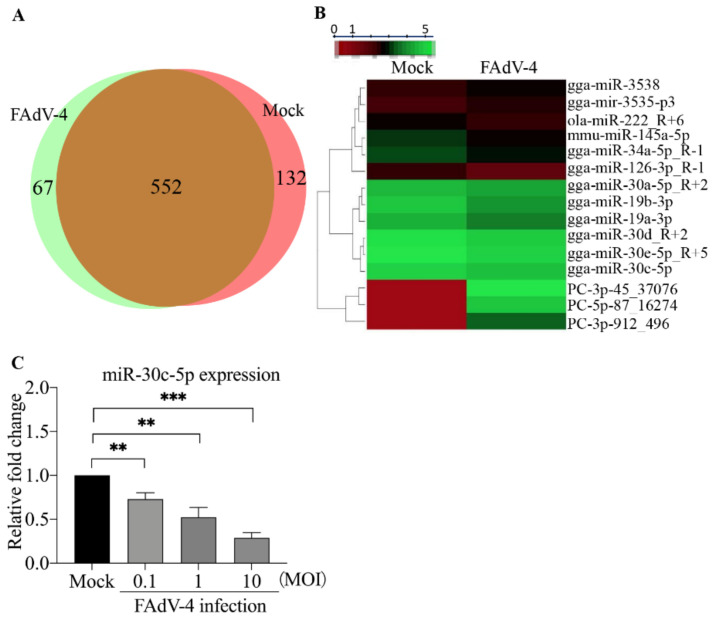
Infection of LMH cells with FAdV-4 strain HuBWH decreases gga-miR-30c-5p expression. (**A**) Venn diagram represents the numbers of miRNAs coded by FAdV-4 and changed miRNAs in LMH cells 24 h after infection with FAdV-4 at an MOI of 1. (**B**) The heat map illustrating the expression profiles of 12 miRNAs in LMH cells with FAdV-4 infection and 3 miRNAs encoded by FAdV-4. The red color stands for the decreased expressions of miRNAs while the green for increased expressions of miRNAs. (**C**) LMH cells were mock-infected or infected with FAdV-4 strain HuBWH at an MOI of 0.1, 1, or 10. Twenty-four hours after FAdV-4 infection, the expression levels of gga-miR-30c-5p were examined by qRT-PCR. The expression of U6 was used as an internal control. The relative levels of gga-miR-30c-5p expression were calculated as follows: gga-miR-30c-5p expression in FAdV-4-infected cells/that of mock controls. Data are representative of three independent experiments and presented as means ± SD. (*** stands for *p* < 0.001; ** for *p* < 0.01).

**Figure 2 viruses-14-00990-f002:**
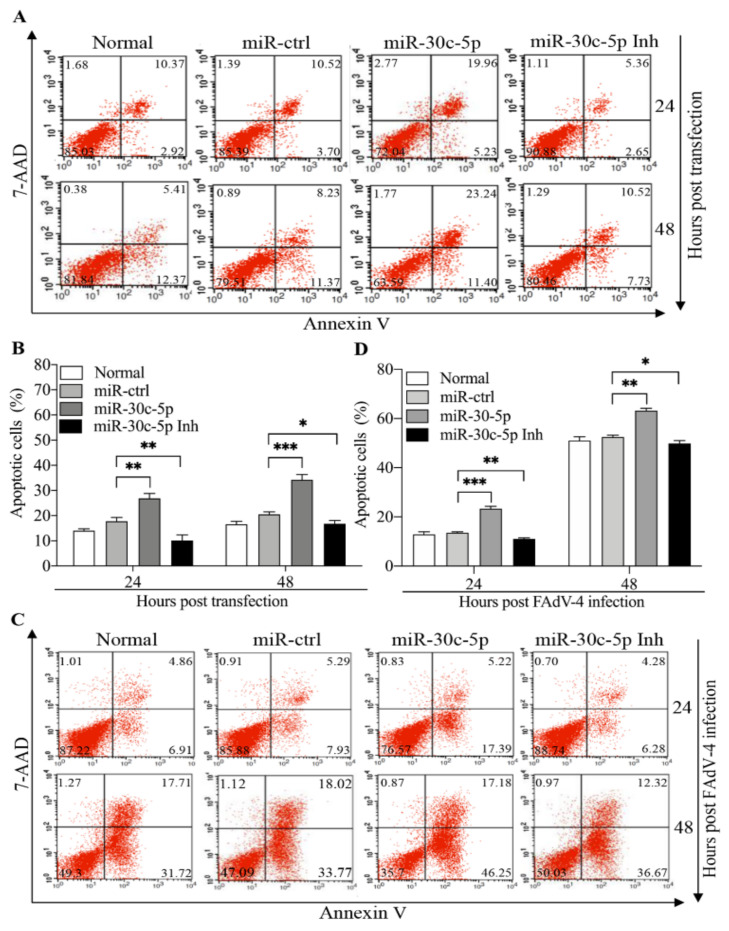
Transfection of LMH cells with gga-miR-30c-5p induces apoptosis and enhances FAdV-4-induced apoptosis. (**A**) The LMH cells were seeded on 12-well plates and cultured overnight, followed by transfection with gga-miR-30c-5p mimics, inhibitors (Inh), or miRNA controls. At different time points (24 and 48 h) post-transfection, cells were harvested, stained with Annexin V-PE and 7-AAD, and examined for apoptosis by flow cytometry. (**B**) The percentage of apoptotic cells in each group shown in panel A was graphed and statistically analyzed. (**C**) The LMH cells were seeded on 12-well plates and cultured overnight, followed by transfection with gga-miR-30c-5p mimics, gga-miR-30c-5p inhibitors or miRNA controls. Twenty-four hours after transfection, cells were infected with FAdV-4 at an MOI of 1. At different time points (24 and 48 h) post-FAdV-4 infection, the cells were examined for apoptosis by flow cytometry as described above. (**D**) The percentage of apoptotic cells in each group shown in panel (**C**) was graphed and statistically analyzed. Data are representative of three independent experiments and presented as means ± SD. (*** stands for *p* < 0.001; ** for *p* < 0.01; * for *p* < 0.05).

**Figure 3 viruses-14-00990-f003:**
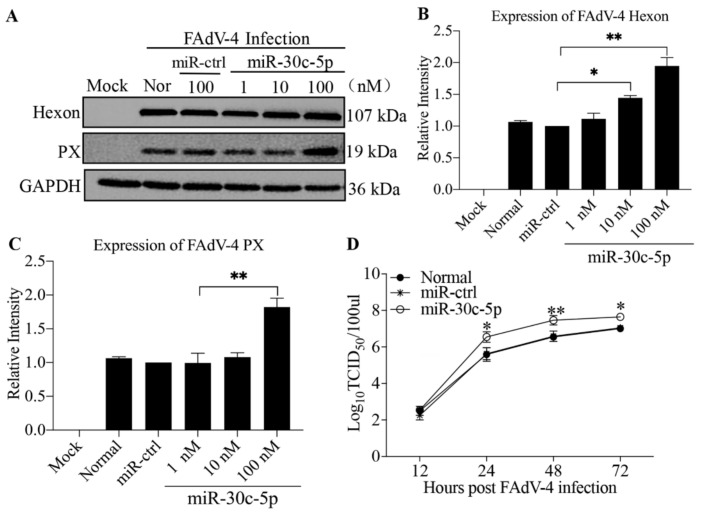
gga-miR-30c-5p facilitates FAdV-4 replication in LMH cells. (**A**–**C**) The LMH cells were transfected with gga-miR-30c-5p mimics or controls at different concentrations. Twenty-four hours after transfection, the cells were infected with FAdV-4 at an MOI of 1. Twenty-four hours after FAdV-4 infection, cell lysates were prepared and subjected to Western Blot analysis using anti-Hexon and anti-PX antibodies. Endogenous GAPDH expression was examined as an internal control. The band densities of Hexon and PX shown in panel (**A**) were quantitated by densitometry, as shown in panels (**B**,**C**). The relative levels of Hexon and PX were calculated as follows: band density of Hexon or PX/that of GAPDH. (**D**) The LMH cells were transfected with gga-miR-30c-5p mimics or miRNA controls. Twenty-four hours after transfection, the cells were infected with FAdV-4 at an MOI of 1. At different time points (12, 24, 48, and 72 h) after FAdV-4 infection, the viral loads in the cell cultures were determined by TCID_50_ assays. The significance of the differences between gga-miR-30c-5p-transfected cells and controls in terms of viral growth was determined by ANOVA. Data are representative of three independent experiments and presented as means ± SD. (** stands for *p* < 0.01; * for *p* < 0.05).

**Figure 4 viruses-14-00990-f004:**
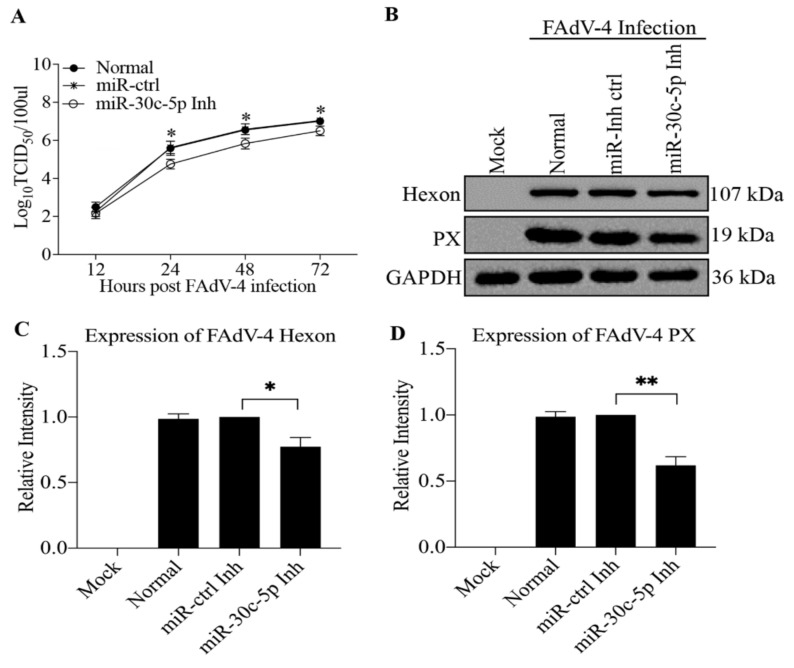
Inhibition of endogenous gga-miR-30c-5p suppresses FAdV-4 replication. (**A**) Knockdown of endogenous gga-miR-30c-5p in LMH cells suppressed FAdV-4 replication. The LMH cells were transfected with gga-miR-30c-5p inhibitor (Inh) or miR-Inh-control (ctrl). Twenty-four hours after transfection, the cells were infected with FAdV-4 at an MOI of 1. At different time points (12, 24, 48, and 72 h) after FAdV-4 infection, the viral loads in the cell cultures were determined by TCID_50_ assays. (**B**–**D**) The LMH cells were transfected with gga-miR-30c-5p Inh or miR-Inh-ctrl. Twenty-four hours after transfection, the cells were infected with FAdV-4 at an MOI of 1. Twenty-four hours after FAdV-4 infection, cell lysates were prepared and subjected to Western Blot analysis using anti-Hexon and anti-PX antibodies. Endogenous GAPDH expression was examined as an internal control. The band densities of Hexon and PX shown in panel (**B**) were quantitated by densitometry, as shown in panels (**C**,**D**). The relative levels of Hexon and PX were calculated as follows: band density of Hexon or PX/that of GAPDH. The significance of the differences between gga-miR-30c-5p Inh-transfected cells and miR-Inh-ctrl in terms of viral growth was determined by ANOVA. Data are representative of three independent experiments and presented as means ± SD. (** stands for *p* < 0.01; * for *p* < 0.05).

**Figure 5 viruses-14-00990-f005:**
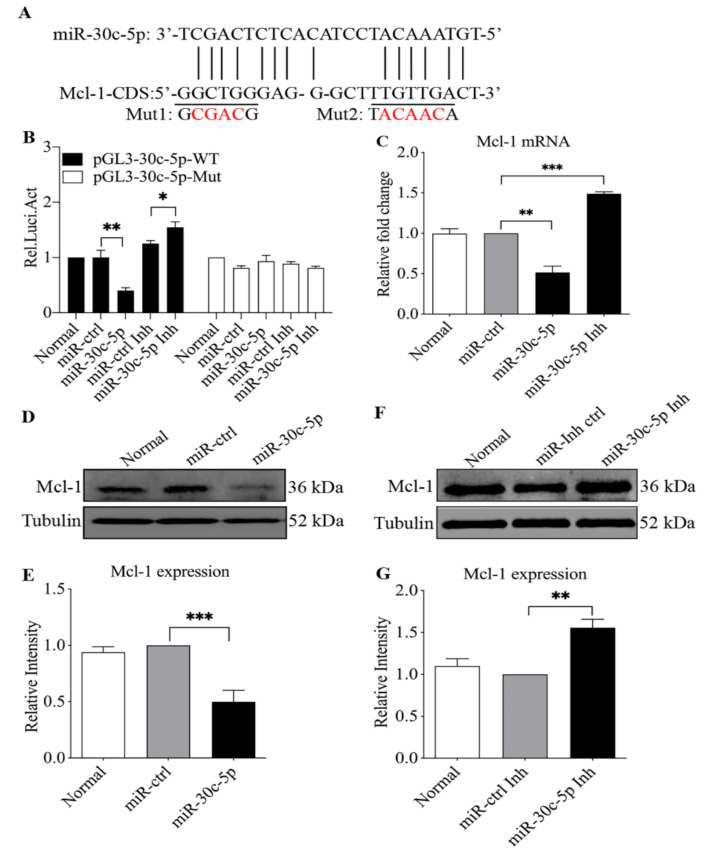
The Mcl-1 gene is a cellular target of gga-miR-30c-5p. (**A**) Diagram of the predicted target site for gga-miR-30c-5p in Mcl-1 gene. The target sequences of gga-miR-30c-5p are underlined and were mutated as indicated. (**B**) The transfection of LMH cells with gga-miR-30c-5p reduced the expression of Mcl-1 but not its mutant. The LMH cells were co-transfected with gga-miR-30c-5p mimics (80 nM), inhibitors (Inh) (200 nM), or miRNA controls, together with luciferase reporter gene vectors (200 ng) and pRL-TK (20 ng). Forty-eight hours post-transfection, the cells were lysed, and a luciferase reporter gene assay was performed to measure Mcl-1 expression. The relative levels of luciferase activity (Rel Luc Act) were calculated as follows: luciferase activity of cells transfected with the reporter gene plasmids together with gga-miR-30c-5p mimics or inhibitors/that of cells co-transfected with the reporter gene plasmids and miRNA controls. (**C**) Effect of gga-miR-30c-5p on Mcl-1 expression. LMH cells were transfected with gga-miR-30c-5p mimics or miRNA controls at 80 nM or gga-miR-30c-5p Inh at 200 nM. Twenty-four hours after transfection, the mRNA expression of Mcl-1 was examined by qRT-PCR. GAPDH was used as an internal control. The relative levels of gene expression were calculated as follows: mRNA expression of Mcl-1 in cells transfected with gga-miR-30c-5p mimics or miRNA inhibitor or miRNA control/that of Mcl-1 in normal cells. (**D**,**E**) The transfection of LMH cells with gga-miR-30c-5p reduced the expression of Mcl-1 protein. The LMH cells were transfected with gga-miR-30c-5p mimics or miRNA controls at 80 nM. Forty-eight hours after transfection, cell lysates were prepared and subjected to Western Blot analysis using anti-Mcl-1 antibodies, and the band densities of Mcl-1 in panel (**D**) were quantitated by densitometry (**E**). Endogenous GAPDH expression was examined as an internal control. The relative levels of Mcl-1 protein were calculated as follows: band density of Mcl-1/that of GAPDH in the same sample. (**F**,**G**) The knockdown of endogenous gga-miR-30c-5p enhanced the expression of Mcl-1 protein. LMH cells were transfected with gga-miR-30c-5p Inh or miR-Inh-ctrl as described in (**C**). Forty-eight hours after transfection, the expression of Mcl-1 protein was examined as described in (**D**), the band densities of Mcl-1 shown in panel (**F**) were quantitated by densitometry (**G**), and the relative levels of Mcl-1 expression were calculated as described in (**E**). Data are representative of three independent experiments and are presented as means ± SD. (*** stands for *p* < 0.001; ** for *p* < 0.01; * for *p* < 0.05).

**Figure 6 viruses-14-00990-f006:**
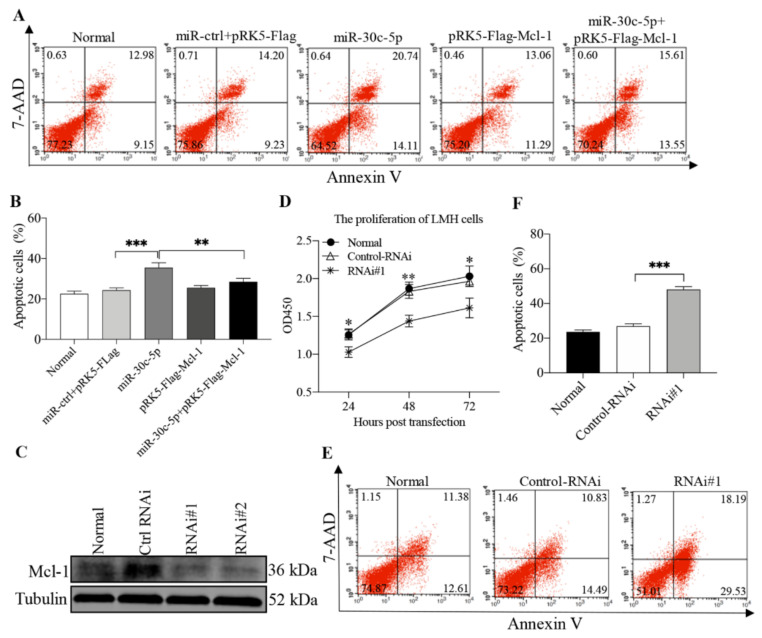
Overexpression of Mcl-1 blocked gga-miR-30c-5p-induced apoptosis in LMH cells, and the knockdown of Mcl-1 promotes apoptosis in LMH cells. (**A**) The LMH cells were co-transfected with gga-miR-30c-5p mimics (80 nM) or miRNA controls (80 nM), pRK5-Flag-Mcl-1(1 μg), or pRK5-Flag (1 μg). Forty-eight hours after transfection, the cells were collected, stained with Annexin V-PE and 7-AAD, and examined for apoptosis by flow cytometry. (**B**) The percentage of apoptotic cells in each group in panel (**A**) was graphed and statistically analyzed. (**C**) Effects of Mcl-1 RNAi on the expression of endogenous Mcl-1. The LMH cells were transfected with siRNA (RNAi#1 and RNAi#2) or controls. Twenty-four hours after the second transfection, cell lysates were prepared and examined by Western Blot using anti-Mcl-1 and anti-Tubulin antibodies. Endogenous Tubulin expression was examined as an internal control. (**D**) The knockdown of Mcl-1 by RNAi suppressed the viability and proliferation of cells. The LMH cells were seeded on a 96-well culture plate and transfected with Mcl-1-RNAi#1 or control-RNAi at a final concentration of 60 nM. At different time points (24, 48, and 72 h) post-transfection, 10 μL of CCK-8 solution was added to each well, followed by incubation at 37 °C for 1 h, and the absorbance of the solution was finally determined at 450 nm using a microplate spectrophotometer. (**E**,**F**) The knockdown of Mcl-1 promotes LMH cells apoptosis. The LMH cells were double transfected with siRNA (RNAi#1) or RNAi controls at an interval of 24 h. Twenty-four hours after the second transfection, the cells were collected, stained with Annexin V-PE and 7-AAD, and examined for apoptosis by flow cytometry. The percentage of apoptotic cells in each group in panel (**D**) was graphed and statistically analyzed (**E**). Data are representative of three independent experiments and presented as means ± SD. (*** stands for *p* < 0.001; ** for *p* < 0.01; * for *p* < 0.05).

**Figure 7 viruses-14-00990-f007:**
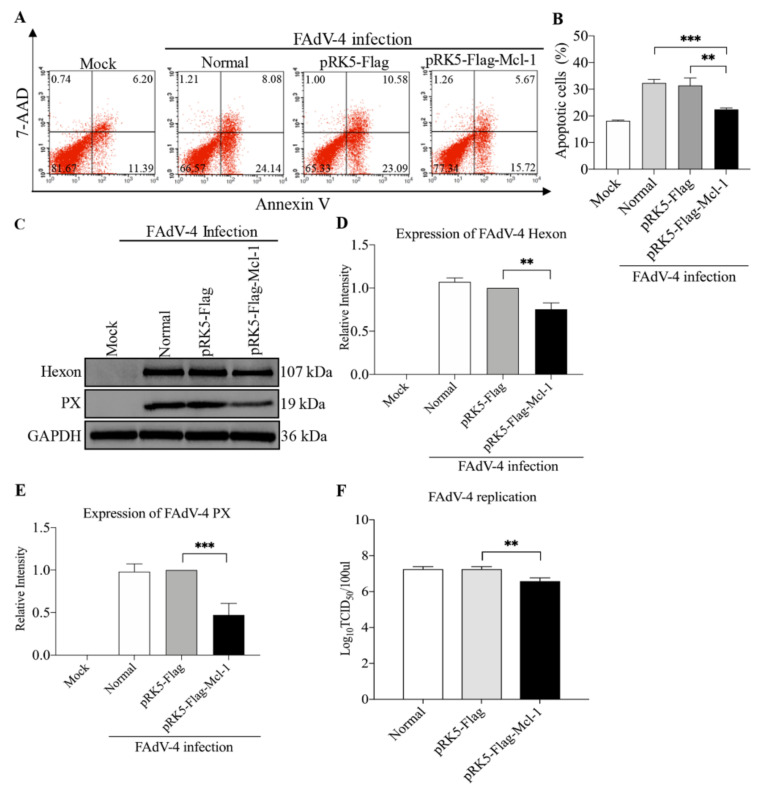
The overexpression of Mcl-1 inhibited FAdV-4-induced apoptosis and suppressed FAdV-4 replication. (**A**,**B**) The overexpression of Mcl-1 inhibited FAdV-4-induced apoptosis. The LMH cells were transfected with pRK5-Flag-Mcl-1 or controls. Twenty-four hours after transfection, the cells were infected with FAdV-4 at an MOI of 1. Forty-eight hours after FAdV-4 infection, the cells were collected, stained with Annexin V-PE and 7-AAD, and examined for apoptosis by flow cytometry (**A**). The percentage of apoptotic cells in each group in panel (**A**) was graphed and statistically analyzed (**B**). (**C**–**E**) The overexpression of Mcl-1 suppressed FAdV-4 replication. LMH cells were treated as described above. Twenty-four hours after transfection, the cells were infected with FAdV-4 at an MOI of 1. Forty-eight hours after FAdV-4 infection, cell lysates were prepared and subjected to Western Blot analysis using anti-Hexon and anti-PX antibodies. Endogenous GAPDH expression was examined as an internal control. The band densities of Hexon and PX in (**C**) were quantitated by densitometry as shown in (**D**,**E**). (**F**) The overexpression of Mcl-1 reduced viral titers in FAdV-4-infected cells. LMH cells were transfected with pRK5-Flag-Mcl-1 vectors or pRK5-Flag as controls, followed by infection with FAdV-4 as described above. Forty-eight hours after FAdV-4 infection, the viral loads in the cell cultures were determined by TCID_50_ assays. Data are representative of three independent experiments and presented as means ± SD. (*** stands for *p* < 0.001; ** for *p* < 0.01).

**Figure 8 viruses-14-00990-f008:**
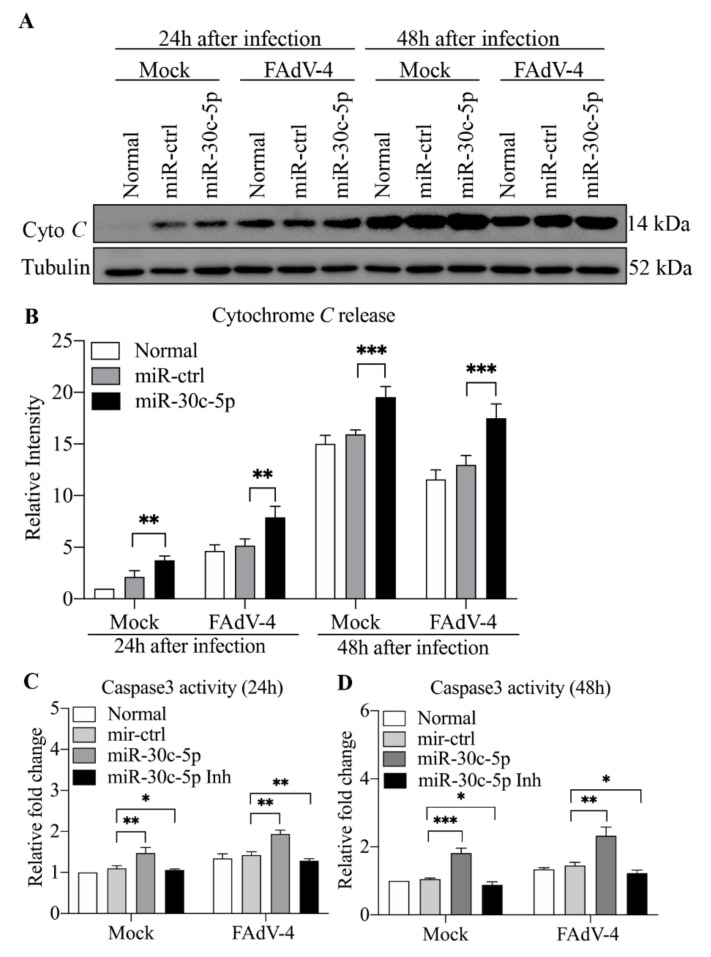
The transfection of cells with gga-miR-30c-5p enhanced FAdV-4-induced cytochrome *c* release and caspase-3 activation. (**A**,**B**) The transfection of cells with gga-miR-30c-5p enhanced the FAdV-4-induced release of cytochrome *c*. The LMH cells were transfected with the indicated miRNAs or miRNA controls. Twenty-four hours after transfection, the cells were mock-infected or infected with FAdV-4 at an MOI of 1. At the indicated time points (24 and 48 h) after FAdV-4 infection, cytosolic proteins were prepared and subjected to Western Blot analysis for the measurement of cytochrome *c* in the cytosol of cells. The band densities of cytochrome *c* in (**A**) were quantitated by densitometry. The relative levels of cytochrome *c* were calculated as follows: band density of cytochrome *c*/that of Tubulin. (**C**,**D**) The transfection of cells with gga-miR-30c-5p enhanced FAdV-4-induced activation of caspase-3. The LMH cells were treated as described above in (**A**). Twenty-four or 48 h after FAdV-4 infection, the activities of caspase-3 were measured at 405 nm with a microplate reader using the substrates DEVD-pNA (synthetic caspase-3 substrate). Data are representative of three independent experiments and presented as means ± SD. (*** stands for *p* < 0.001; ** for *p* < 0.01; * for *p* < 0.05).

**Table 1 viruses-14-00990-t001:** The primers of PCR for Mcl-1sequences.

	Sequences (5′–3′)
Mcl-1 sense primer	AGGACGACGATGACAAGGGATCCATGTTTGCAGTCAAGCGGA
Mcl-1 antisense primer	CGGCCAAGCTTCTGCAGGTCGACTCACCGGATCATGTAGGCCAAGCTC

**Table 2 viruses-14-00990-t002:** Sheet of miRNAs sequences.

The Name of miRNA	Sequences (5′–3′)
gga-miR-30c-5p mimics	UGUAAACAUCCUACACUCUCAGCU
mimics negative control	UUCUCCGAACGUGUCACGUTT
gga-miR-30c-5p inhibitors	AGCUGAGAGUGUAGGAUGUUUACA
inhibitors negative control	CAGUACUUUUGUGUAGUACAA

**Table 3 viruses-14-00990-t003:** The primers of qRT-PCR for Mcl-1 and GAPDH sequences.

The Name of Gene	Sense Primers (5′–3′)	Antisense Primers (5′–3′)
Mcl-1	GGGATCATCACGGACGCATTGG	TCCTCAACTCGGAAGAAGTCAACAAAG
GAPDH	CAACTACATGGTTTACATGTTCC	GGACTGTGGTCATGAGTCCT

**Table 4 viruses-14-00990-t004:** Sheet of siRNAs sequences.

The Name of siRNA	Sense Primers (5′–3′)	Antisense Primers (5′–3′)
RNAi#1	CUCAUCUCAUUUGGUGCCUTT	AGGCACCAAAUGAGAUGAGTT
RNAi#2	GCCUACAUGAUCCGAAAGUTT	ACUUUCGGAUCAUGUAGGCTT
negative siRNA control	UUCUCCGAACGUGUCACGUTT	ACGUGACACGUUCGGAGAATT

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
