# Peer review of "Gga-miR-30c-5p Enhances Apoptosis in Fowl Adenovirus Serotype 4-Infected Leghorn Male Hepatocellular Cells and Facilitates Viral Replication through Myeloid Cell Leukemia-1"

_viruses, 2022, doi:10.3390/v14050990_

Round 1
Reviewer 1 Report
The authors studied the apoptotic effect of gga-miR-30c-5p microRNA in the Fowl adenovirus 4 (FAD-4) affected hepatocellular cells with the target to know the effect on viral replication. The authors also claimed that it unraveled the ongoing research on the viral replication mechanism in the host. It is a good research study in the progress of knowledge of the microRNA effects of carcinogenesis
- Title can be changed to “Gga-miR-30c-5p Enhances Apoptosis in Fowl Adenoviruses 4-infected Leghorn Male Hepatocellular Cells and Facilitates Viral Replication through Myeloid Cell Leukaemia-1.
- Abstract:
Line 15: 'pulled up' or 'evidenced' can be used instead of "attracted much attention".
- Line 16: 'have shown' or 'showed' is better to use instead of 'show'.
- Introduction, Line 44: 'attracted' should be replaced with 'pulled up' or 'evidenced' as well as in further use.
- Line 51: Correct spacing between 11 and structural.
- line 76: “…serves as a pro-apoptosis factor…” should be “…serves as a pro-apoptotic factor…”
- Material and methods, Line 102-139: Several primers have been designed to amplify target sequences. It is better to represent all the primer sequences in a table for a better understanding by the readers.
- page 3, line 122: “RNA misi microRNA kit” should be “RNA mini microRNA kit”
- Material and methods, page 4, line 173: “Western Bolt” should be corrected “Western Blot”
- Results, page 5, line 228 : “the roles of miRNAs play in host response”. It is better to remove “play”
- Results, page 13, line 144 : “acts as an anti-apoptosis factor” should be “acts as an anti-apoptotic factor”
- Results : Data on cells viability and cytotoxicity at used concentrations of siRNAs, miRNAs, and inhibitors should be provided.
- Figures : the molecular weights of detected proteins should be indicated in western blots figures (Fig.3-8).
- Discussion, page 17, line 553: “….all these virus (IBDV, Reovirus, and FAdV) are non-enveloped virus….” should be “….all these viruses (IBDV, Reovirus, and FAdV) are non-enveloped viruses….”
Reviewer 2 Report
Haiyilati et al report an analysis of changes in miRNA expression during infection of an immortalized chicken liver cell line (LMH cells) with fowl adenovirus 4 (FAdV-4). FAdV-4 is an important virus to study currently because it is the causative agent of HHS syndrome in chickens, a economically costly disease.
They observe that most miRNAs are differentially expressed during infection and that a number of viral encoded miRNAs are also detected, which is interesting. They focus on the downregulation of gga-miR-30c-5p, because of its likely role in apoptosis. They show that gga-miR-30c-5p is downregulated in a dose dependent fashion by FAdV-4. Using a variety of overexpression or inhibition based assays, serves as a pro-apoptotic factor, facilitates viral gene expression and progeny production, negatively impacts expression of the Mcl-1 anti-apoptotic Bcl-2 family member. gga-miR-30c-5p effects on Mcl-1 are likely mediated via a predicted target site that is mapped and studied in the context of a luciferase report assay with or without a functional target site. They also show that Mcl-2 suppresses gga-miR-30c-5p and FAdV-4 induced apoptosis.
this is a generally well-written, well-planned and well-controlled study with lots of experimental data. Although most effects of Gga-miR-30c-5p are rather small, they are statistically significant and the results are mutually reinforcing enough that I am convinced that their interpretation is sound.
No major points that need repair from my perspective.
Minor points:
Line 16 - unraveled is a poor word choice here. Use unresolved, unknown or not fully understood?
Line 51 - missing space between 11 and structural?
Fig. 1: panel A - the legend and results do not describe this panel the same way and as presented its hard to interpret. Looking at this figure, I would suspect that cellular 132 miRNAs are unchanged by infection, 552 are significantly changed and 67 viral miRNA are detected?
Fig. 1: panel B. The heat map is uninformative with that color scale. gga-miR-30c-5p expression differences are so small that the same shade of green is present in the heatmap. Change the color transition choices in the software to show a meaningful change in color for differentially expressed miRNA. As presented, looking at this panel, there is no obvious expression difference to justify pursuing gga-miR-30c-5p studies. It just sounds and looks weak.
Line 265 - processed is a poor word choice. Use proceeded?
I think that it would add a bit of value to the paper if the authors did a quick analysis to see if the gga-miR-30c-5p target they identify in chicken Mcl-1 is present in all other avian species likely to support FAdV-4 infection? If the miRNA target is conserved, this would strengthen their conclusions slightly.
Finally, its a bit counter-intuitive that gga-miR-30c-5p is reduced during infection but functions to promote infection when overexpressed. What is a proposed explanation for why FAdV-4 downregulates an target that promotes virus replication and yield? This should be mentioned in the discussion explicitly.
Round 2
Reviewer 1 Report
- The manuscript needs a significant English revision.
- Abstract line 17, Introduction line 47: “pulled up much attention” should be “pulled up”.
- Material and methods, page 3, line 116: “with primers as showing in Table 1” should be “with primer pairs shown in Table 1”.
- Material and methods, page 3, line 122: “The sense sequences for miRNAs are as follows” should be “The sense sequences for miRNAs are shown in Table 2”
- Material and methods, page 4, line 138: “Primers used for…were as showing in Table 3” should be “Primers used for…. are shown in Table 3”
- Material and methods, page 5, line 222: “showing table 4” should be “are shown in Table 4”
- Table 3 : “The names of genes” should be “the name of gene”
- Table 4 : “The names of siRNAs” should be “The name of siRNA”
- Material and methods, page 5, line 177: “Normal cells cells receiving” should be “Normal cells receiving”
- The results of cell viability upon siRNAs or miRNAs transfection should be included in the manuscript.
- In some figures, the molecular weights of proteins are not indicated.
Author Response
Point-by-point response to comments from reviewers
The manuscript needs a significant English revision.
- Abstract line 17, Introduction line 47: “pulled up much attention” should be “pulled up”.
Reply:Thank you very much for the suggestion. We have deleted the word “much” in Abstract and Introduction in the revised MS as suggested. (Please check line 16 and 46 in the revised MS).
- Material and methods, page 3, line 116: “with primers as showing in Table 1” should be “with primer pairs shown in Table 1”.
Reply:Thank you very much for the suggestion. We have changed “with primers as showing in Table 1” to “with primer pairs shown in Table 1” in the revised MS as suggested. (Please see line 107 in the revised MS).
- Material and methods, page 3, line 122: “The sense sequences for miRNAs are as follows” should be “The sense sequences for miRNAs are shown in Table 2”
Reply:Thank you very much for the suggestion. We have changed the sentence “The sense sequences for miRNAs are as follows” to “The sense sequences for miRNAs are shown in Table 2” in the revised MS as suggested. (Please see line 113 in the revised MS).
- Material and methods, page 4, line 138: “Primers used for…were as showing in Table 3” should be “Primers used for…. are shown in Table 3”.
Reply:Thank you very much for the suggestion. We have changed the sentence “Primers used for qRT-PCR were as showing in Table 3” to “Primers used for qRT-PCR are shown in Table 3” in the revised MS as suggested. (Please see line 124 in the revised MS).
- Material and methods, page 5, line 222: “showing table 4” should be “are shown in Table 4”.
Reply:Thank you very much for the suggestion. We have changed “showing Table 4” to “are shown in Table 4” in the revised MS as suggested. (Please see line 192 in the revised MS).
- Table 3 : “The names of genes” should be “the name of gene”
Reply:Thank you very much for the suggestion. We have changed “The names of genes” to “The name of gene” in the revised MS as suggested. (Please see Table 3 in the revised MS).
- Table 4 : “The names of siRNAs” should be “The name of siRNA”
Reply:Thank you very much for the suggestion. We have changed “The names of siRNAs” to “The name of siRNA” in the revised MS as suggested. (Please see Table 4 in the revised MS).
- Material and methods, page 5, line 177: “Normal cells cells receiving” should be “Normal cells receiving”
Reply:Thank you very much for the comment. We apologized for missing the word “or” between the two “cells”. We have revised “Normal cells cells receiving” for “Normal cells or cells receiving---”in the revised MS. (Please check line 155 in the revised MS).
- The results of cell viability upon siRNAs or miRNAs transfection should be included in the manuscript.
Reply:Thank you very much for the suggestion. We have added the result of cell viability upon siRNA transfection as Figure 6D to the revised MS. As the Figure with miRNAs transfection contains too many panels, it is too difficult to fit the result of cell viability upon miRNAs transfection into the figure with miRNAs transfection. Thus, we did not change this figure. If the addition of the cell viability upon miRNAs transfection is a must, we could add it the revised MS as a supplemental figure. Thank you very much for your understanding and consideration.
- In some figures, the molecular weights of proteins are not indicated.
Reply:Thank you very much for the comment. We have indicated the molecular weights of detected proteins in all Western Blot figures in the revised MS as suggested. (Please see Figures 3-8 in the revised MS).
We really appreciated the constructive comments by the reviewer, which are important to improve the quality of our revised manuscript.
Round 3
Reviewer 1 Report
Gga-miR-30c-5p Enhances Apoptosis in Fowl Adenoviruses 4- infected Leghorn Male Hepatocellular Cells and Facilitates Viral Replication through Myeloid Cell Leukaemia-1
Comments :
There are still some flaws in the manuscript, as followings. Please check again carefully and ask preferably English edition to native speaker.
- Title of the manuscript: “Fowl Adenoviruses 4” should be “Fowl Adenovirus 4”
- Abstract, line 14: “Fowl adenoviruses 4 (FAdV-4) is….” should be “Fowl adenovirus 4 (FAdV-4) is….”
- Abstract, line 25: “Fad5V-4” should be “FAdV-4”
- Introduction, page 2, line 63: “posttranscriptional” should be “post-transcriptional”
- Material and methods, page 4, line 159 : “To confirm the effect of Mcl-1 on FAdV-4 replication. LMH cells were….”. Replace point to comma.
- Material and methods, page 4, line 162 : “50% tissue culture infective doses” should be “50% tissue culture infectious dose”
- Figure 3-B title : “Experssion of FAdV-4 Hexon”. Please correct “Expression”
- Discussion, page 17, line 567: “…all these virus…” should be “…all these viruses…”